# Study on Crack Development in Red Clay from Guangxi Guilin with Different Clay Grain Content

**Baochen Liu** [1,†], **Liangyu Wang** [2,†] **and Bai Yang** [1,*]

1   School of Architecture and Transportation Engineering, Guilin University of Electronic Technology, Guilin 541004, China

2   School of Civil and Architectural Engineering, Guilin University of Technology, Guilin 541004, China

*   Correspondence: ayangbai@163.com; Tel.: +86-0773-2303796

†   These authors contributed equally to this work and should be considered co-first authors.

**Abstract:** In order to study the influence of different clay contents on the fractality of red clay, specimens having four different water contents were prepared. The cracking characteristics of the specimens were observed at 20 °C and 60 °C. Image J software was used to measure and calculate the crack area, crack ratio, crack length and width of each sample. The test results showed that the development of cracks in red clay could be divided into three stages: crack generation, crack development and crack stabilization. The clay particle content, temperature and water content have significant effects on crack development, and from the test analyses, it was determined that for construction in the Guilin area, it is necessary to pay attention to drainage protection.

**Keywords:** red clay; clay content; crack; temperature; moisture content

## 1. Introduction

Red clay is a special kind of soil that exhibits surface shrinkage and water loss cracking [1]. The Guilin region has high temperatures and abundant rain in the summer, while the winter is dry and cold. The effects of climate are strong, and the soil easily shrinks and cracks. Soil dry shrinkage and cracking can be attributed to external and internal factors. External factors include light, temperature, humidity, air flow, etc., which directly affect the energy supply that is involved in the process of water evaporation from the soil body; this has a significant impact on the rate of water evaporation, thus affecting crack development in the soil body. Internal factors are mainly composed of the physical properties and structure of the soil body itself, which have a significant impact on its adsorption capacity of bound water, and on the migration channels of free water in the soil body [2–6]; as the mechanical properties of the soil body change, the development process of cracks also changes [7–10]. The dry shrinkage and cracking of soils can cause geological disasters such as landslides and road cracking. Consequently, many scholars have studied the generation of fissures on the basis of temperature, humidity, soil thickness, hydraulic gradient, and wet and dry cycles [11–17]. Observations of the fissure development process are also very important to the process of conducting experimental research on the fissure development laws of soil. With the continuous progress of science and technology, many scholars have begun to use various types of software and experimental devices to observe fissures more accurately [18–27]. Although the clay content in red clay from the Guilin area is highly variable and unevenly distributed, the effect of clay content on fissures has not yet been studied by scholars. In this paper, we took Guilin red clay soil as the research object, prepared red clay soil specimens with different viscous particle contents at the optimum, plastic, natural and liquid limits of water content, and subjected them to two temperature conditions, 20 °C and 60 °C (from the monitoring of surface temperatures in Guilin, the maximum temperature is 20 °C in winter and 60 °C in summer). We took photos to observe the processes of water loss and cracking in the soil specimens, and used

Photoshop to process the images The crack area, crack ratio, crack length, crack width and surface shrinkage were measured and calculated using Image J image processing software (National Institutes of health. Bethesda, MD, USA), in order to analyze the crack change patterns in the soil specimens.

## 2. Test Principles and Scheme

### 2.1. Determination of Basic Physical Properties

The soil parameters are shown in Table 1.

**Table 1.** The soil parameters.

| Natural Moisture Content/(%) | Natural Density/(g/cm³) | Specific Gravity of Solid Particles | Optimum Moisture Content/(%) | Maximum Dry Density/(cm³) | Liquid Limit/(%) | Plastic Limit/(%) |
|---|---|---|---|---|---|---|
| 41 | 1.8 | 2.76 | 24 | 1.55 | 56.5 | 31.8 |

### 2.2. Preparation of Viscous Pellets

Particles with sizes $\leq 0.005$ mm are clay particles, and clay particles required for testing were prepared using the net water sedimentation principle; the preparation process is shown in Figure 1a. As a result of the small volume of the ring-knife soil specimen and the inconspicuous fissure development pattern, a homemade 210-millimeter diameter compactor was used to prepare the specimen, as shown in Figure 1b. The clay content of the original soil specimen in this test was 45.39%. Then, compacted specimens with clay contents of 55.39%, 65.39% and 75.39% were prepared. The prepared specimens were dried in the oven at 20 °C and 60 °C, photographed and weighed, and the photographed and weighed equipment is shown in Figure 1c.

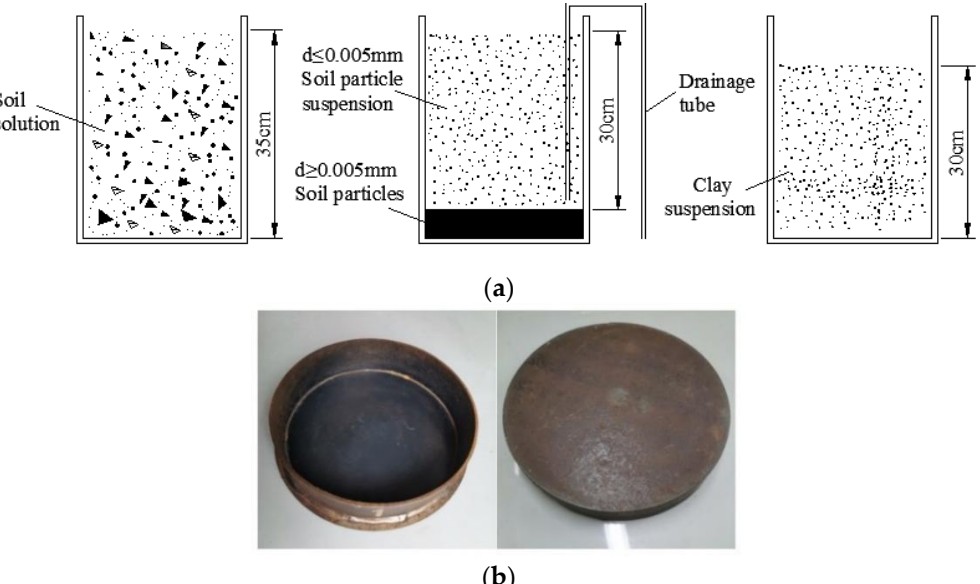

(a)

(b)

**Figure 1.** *Cont.*

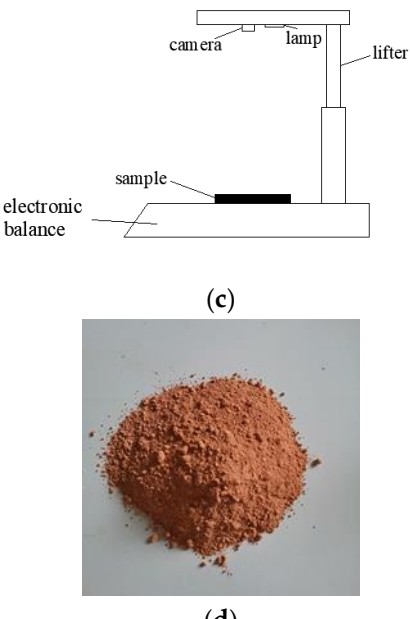

**(c)**

**(d)**

**Figure 1.** Test equipment. (**a**) Schematic diagram of viscous particle extraction; (**b**) compressor; (**c**) schematic diagram of the photographic weighing equipment; (**d**) viscous particles.

The following describes the extraction process of the clay particles:

(1)  The soil was weighed and soaked

A total of 1.0 kg of soil was used; The soil to water ratio was 1:6. The soil was soaked in water for 24 h.

(2)  The sample was heated to boiling

Water was added to dilute the mud so that the ratio of soil to water reached 1:10; then, the mud was heated to boiling and stirred for 1 h.

(3)  Net water sedimentation was determined

A total of 100 mL of sodium hexametaphosphate with a concentration of 4% was added to the cooled mud. Then, 20 L net water was added and stirred into the mud. Timing began after stirring was stopped. According to the principle of net water sedimentation, the time for the sedimentation of soil particles with a particle size of 0.005 mm was calculated as follows:

$$t = \frac{K^2 L_t}{d^2} \tag{1}$$

where $t$ is the time for sedimentation (s); $d$ is the particle size (mm); $K$ is the calculation coefficient of particle size that is related to the temperature of the suspension and to the specific gravity of the soil particles, executed according to "Standards for Geotechnical Test Methods" (GB/T50123-2019). $L_t$ is the distance that the soil particles settle in a certain time $t$. $L_t = a - b(R_2 - 1)$ (cm), and the parameters of the density meter used in this experiment were a = 20.01 and b = 646.333.

It can be seen from Equation (1) that after calculating the time $t$, the soil particles with a particle size greater than or equal to $d$ will settle below the falling distance $L_t$. Soil particles with a particle size smaller than $d$ required for the test are in suspension above the depth of $L_t$.

(4)  Clay particles were extracted

After extracting the suspension, it was filtered and rinsed with water to remove the sodium hexametaphosphate. The washed clay particles were dried in an oven and crushed for use. The prepared viscous particles are shown in Figure 1d.

### 2.3. Test Procedure

(1)　Soil specimens were prepared with different moisture contents.

(2)　The compactor specimen discs were numbered and weighed, and records were made. Then, the soil specimen with the configured moisture content was poured into a compactor that was coated with Vaseline; the specimen was compacted, weighed and photographed together with the disc, then placed in a constant temperature oven. Cracking of the soil specimen is mainly caused by the shrinkage of the soil mass due to water loss. The binding effect of the disc periphery on the soil specimen is limited, but the binding effect of the disc bottom on the soil specimen is significant. In order to reduce the binding effect of the disc bottom on the soil specimen, we used a layer of Vaseline to coated the disc bottom and periphery, which reduced the impact of the disc surface friction.

(3)　In the process of drying, the specimens were photographed and weighed every 4 h at 20 °C, and every 0.5 h at 60 °C. When the quality of the specimen no longer changed, testing was considered complete.

### 2.4. Principles of Crack Development

The development of cracks in red clay can be divided into a water loss process and a cracking process. Water loss in the soil mainly involves the process of water in the soil absorbing the energy in the atmosphere, and converting it into a gaseous state to be dissipated into the air. The water molecules in the soil body consist of free water and bonded water. Free water diffuses to the atmosphere after absorbing energy; bonded water needs to absorb more energy to break free because it is combined with minerals in the soil body; the loss of bonded water will affect the crystal layer spacing of some mineral components, which causes contraction of the soil body. In the process of soil free water dissipation, pore water decreases, air enters the void, the pore water surface curvature increases, surface tension increases, matrix suction increases, the surrounding soil body is pulled and squeezed into the void, the soil particles between the void decrease, thus soil body volume contraction occurs. When the free water is dissipated, the water in the soil body that is bound by the electrostatic gravitational force of the soil grain minerals also begins to dissipate gradually. In the lattice structure of soil grain minerals, the thickness of the hydration film formed by the bound water between the crystal layers determines the distance between adjacent cells. When the bound water in the soil grain minerals gradually decreases, the hydration film becomes thinner, the distance between cells in the soil grain minerals decreases, and the soil body further undergoes volume contraction. In the process of soil water loss and contraction, water evaporates from the upper part of the soil body before the lower part of the soil body, and the contraction direction and contraction distance of the soil body change during the overall contraction process. Thus, the soil body produces horizontal tensile stress inside the soil body, and when the tensile stress is greater than the tensile strength of the soil body, the soil body pulls apart, and cracks are produced on the surface. As the water continues to dissipate, the soil shrinks further, and the resulting fissures become increasingly wide.

## 3. Experimental Results

### 3.1. Crack Development Map

Figures 2 and 3 show the beginning and end of fissure development for different moisture contents in red clay specimens, with 45.39%, 55.39%, 65.39% and 75.39% clay content at 20 °C and 60 °C.

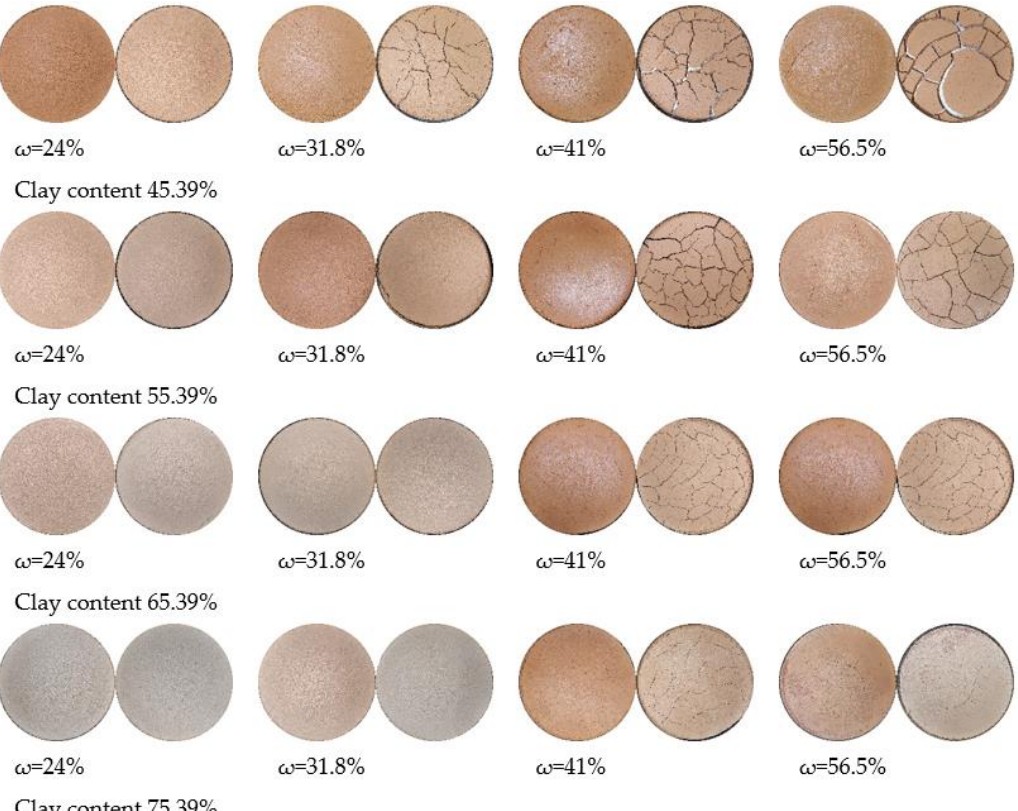

**Figure 2.** Crack development diagram for 20 °C.

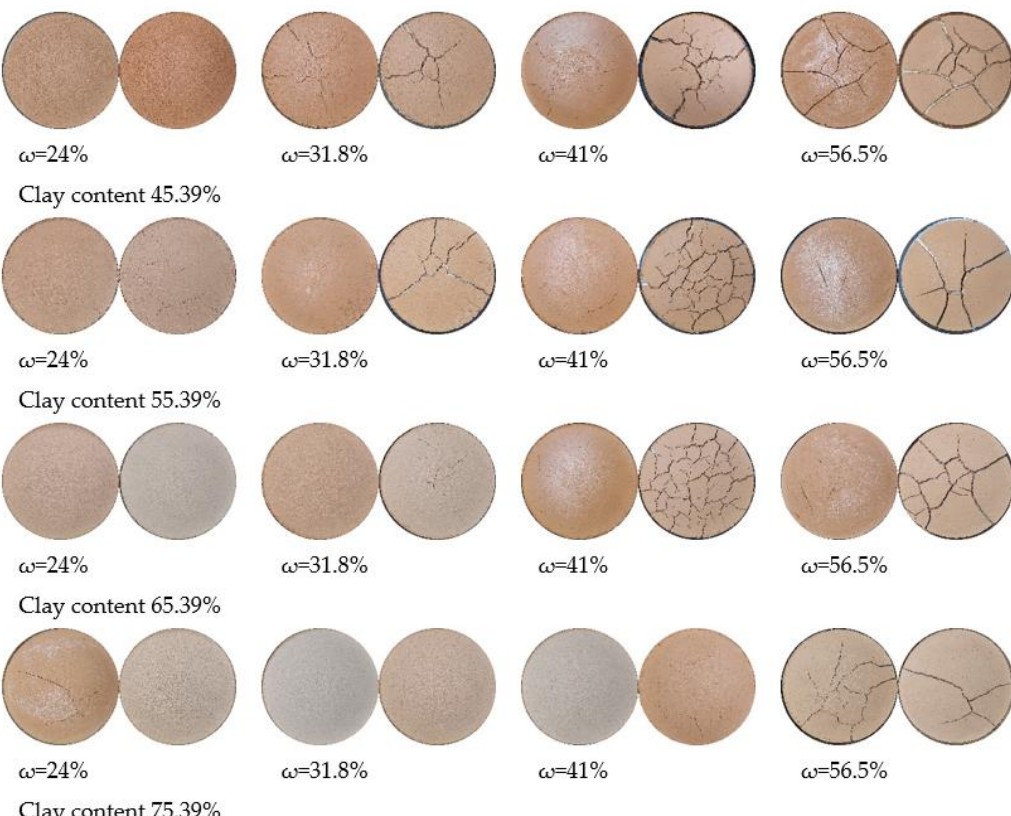

**Figure 3.** Crack development diagram for 60 °C.

### 3.2. Red Clay Surface Crack Treatment

The crack images were processed in Photoshop in order to quantify relevant data for the cracks; then, the crack area, length and width were measured and calculated using Image J software, and the processed images are shown in Figure 4 and Table 1. Mechanical properties of the grouting material

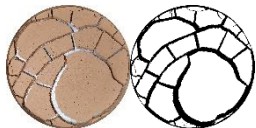

**Figure 4.** Binary image processing.

## 4. Analysis of Test Results

### 4.1. Effect of Temperature on Crack Area

The crack area refers to the horizontal area of the specimen crack, which was obtained by processing the specimen surface with the image area measurement function of Image J software. The units of area used were $mm^2$. It can be observed from the crack development diagram that the temperature had a significant effect on the crack area. The graphs of the crack area with time for red clay specimens with different viscous particle contents at 20 °C and 60 °C environments are shown in Figure 5.

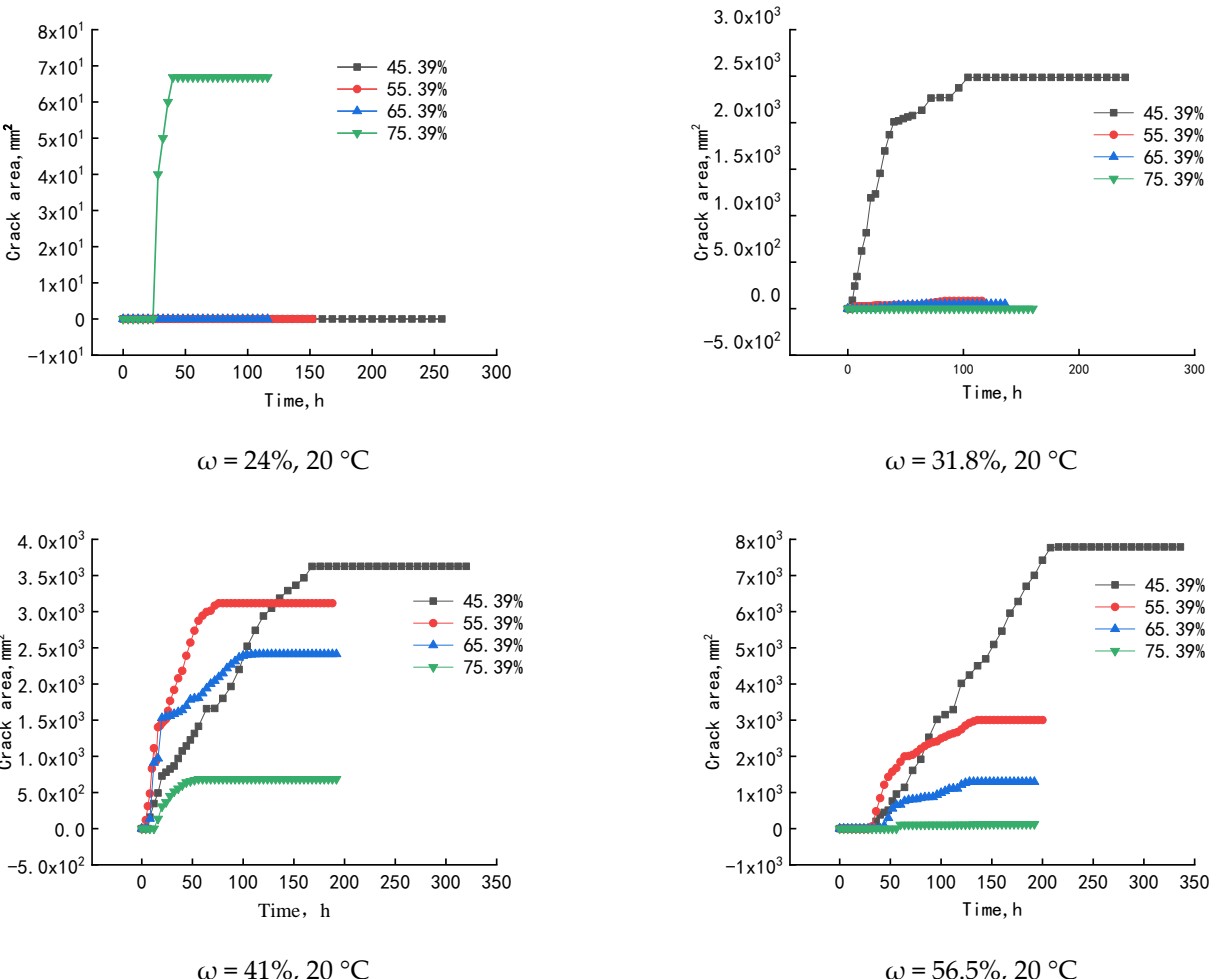

ω = 24%, 20 °C

ω = 31.8%, 20 °C

ω = 41%, 20 °C

ω = 56.5%, 20 °C

**Figure 5.** *Cont.*

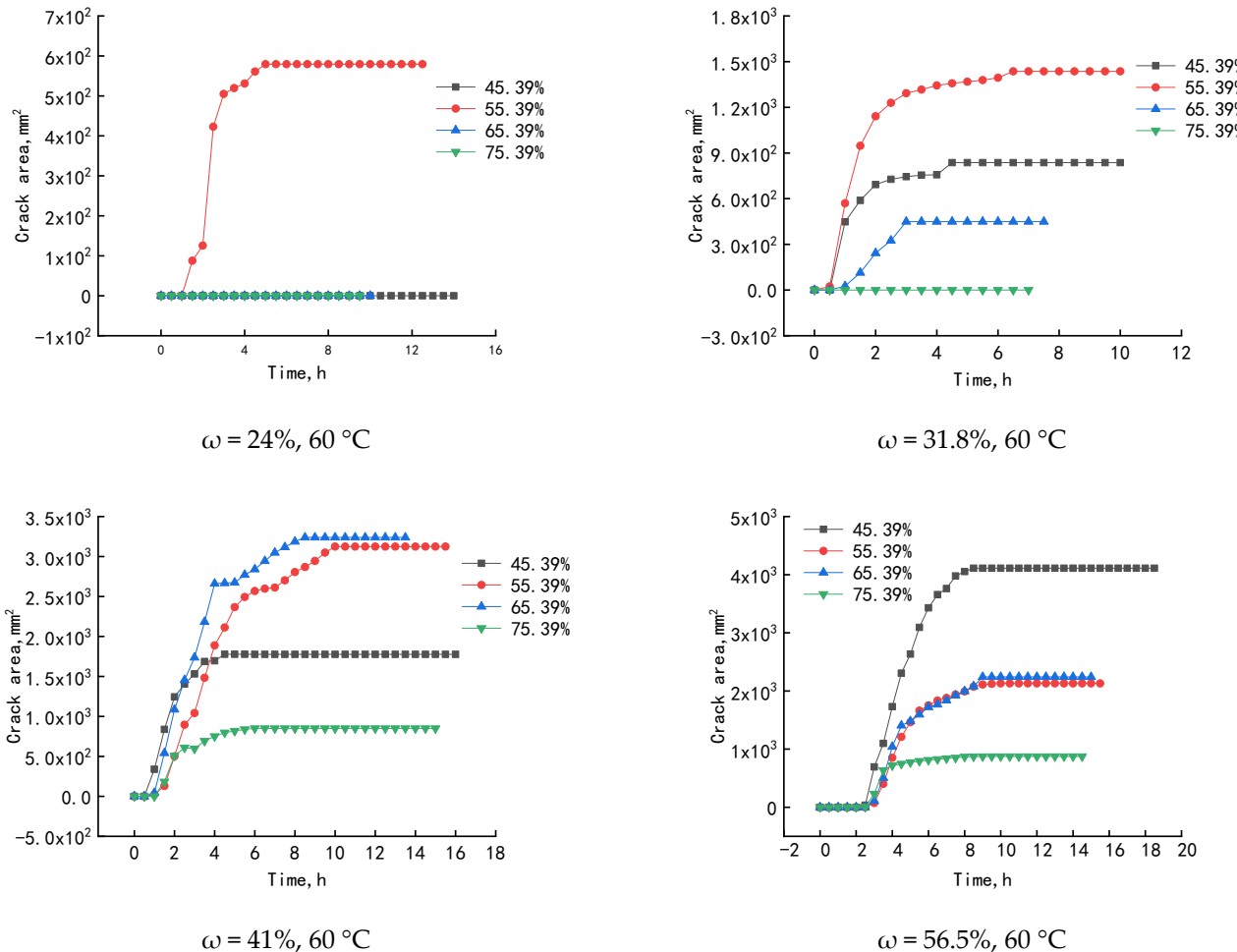

**Figure 5.** Variation curves of crack area with time for specimens with different initial water content and different viscous particle content.

From Figure 6, when ω = 24%, only the specimen with 55.39% mucilage content at 60 °C and the specimen with 75.39% mucilage content at 20 °C showed a small number of fissures; the fissure rate was only 1.67% and 0.19%, and temperature had no significant effect on fissure development in the specimen with the optimum initial water content. When the initial water content of the specimens remained unchanged, the crack rate of the specimens with 45.39% mucilage content at 60 °C showed a significant decrease compared with that at 20 °C; this was due to the rapid rate of evaporation and water loss in the specimens at 60 °C, and to rapid shrinkage of the soil. The specimens with 55.39%, 65.39% and 75.39% clay content showed an increase and decrease in the crack rate with an increase in temperature, when the initial water content was constant. With an increase in temperature, the water loss rate of the soil body increased significantly, and the shrinkage rate of the soil body also accelerated. The bound water in the soil particles is more likely to absorb enough energy to dissipate, so that shrinkage of the soil body increases, and damage to the soil body caused by the internal forces of shrinkage also increases; hence, the crack rate increases.

With increases in the clay content, the final crack area of specimens with the same initial water content showed a decreasing trend. The effect of clay content on the final crack area of the specimens is due to the change in soil particle gradation. The particle gradation of the soil with low clay content was better than that of the soil with high clay content, and the soil structure was more compacted during the compaction of the specimen. With gradual increases in viscous particle content, the particle gradation of soil tends to be poor, and the contact surface of fine particles in the compaction process facilitates the production of pores. The volume of the pores is small, but their numbers increase, which causes water

loss; the shrinkage occurs mostly around these pores, thus the internal pores of the soil become larger, and the porosity increases. When the shrinkage of the soil around the pores becomes too large, the pores are pulled and cracked, resulting in fissures.

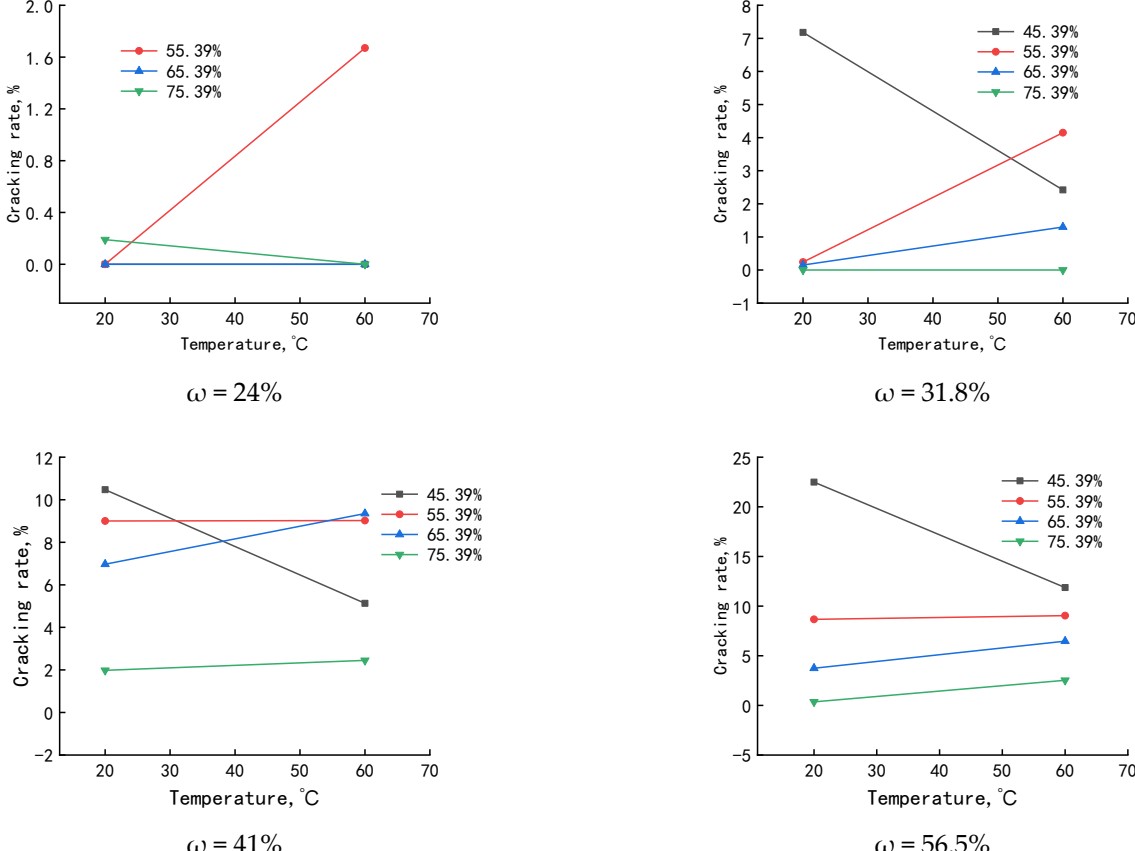

**Figure 6.** Variations in cracking rate with temperature.

### 4.2. Effect of Temperature on Crack Length

The crack length refers to the length of the crack centerline in the crack image on the specimen surface, which is obtained by processing with Image J software. The units of length unit used were mm. The lengths of cracks affected by temperature for specimens with different viscous particle contents at the same initial water content are shown in Figure 7.

It can be seen from Figure 7 that the crack length in specimens with the same viscous particle content at the same initial moisture content varied significantly with temperature, but did not show regularities. There is no correlation between the crack length and the variation in mucilage content of the specimens. The development of the crack length is related to the water loss and shrinkage of the specimen, and the internal structure of the soil; however, the specimen used in this test was a heavy plastic compacted specimen, and the soil was an inhomogeneous medium. The internal structure of each specimen was different, and the temperature could not influence the initial soil structure of the specimen; thus, no clear correlation was found between the development of crack length in the specimen and the ambient temperature.

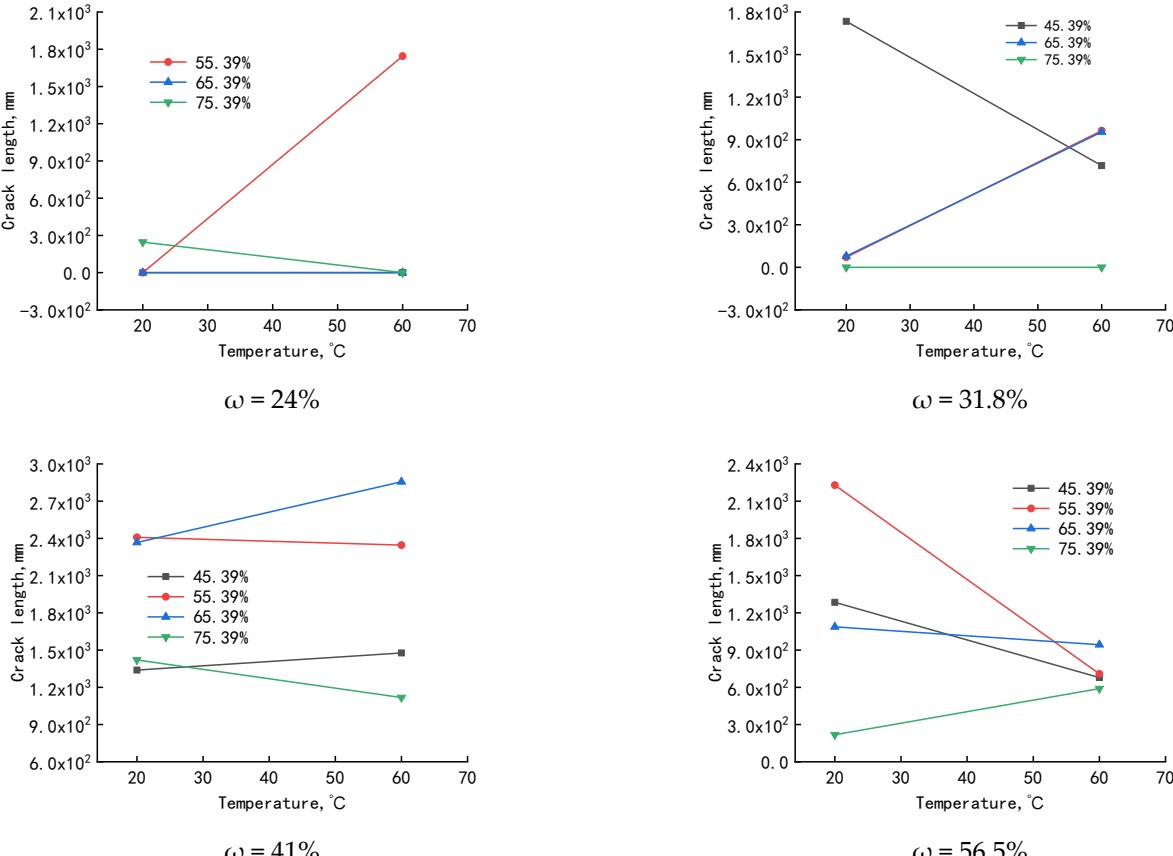

**Figure 7.** Changes in crack length with temperature.

### 4.3. Effects of Temperature on Crack Width

The average width was adopted for the statistics of crack width. It was obtained by dividing the crack area by the crack length. The units used for width were mm. For the analysis of crack development, since the crack width was not fixed and the crack width varied in different locations, the average width was used in the width calculation, which was obtained by the ratio of crack area and crack length, as shown in Table 2.

Analysis of the data in Table 2 shows that the effect of temperature on the average crack width is significant when the initial water content is the same. The effects of temperature on the average fissue width are shown in Figure 8.

It can be seen from Figure 8 that there is no obvious pattern of fissure width when $\omega = 24\%$ and when $\omega = 31.8\%$. The average widths of fissures of the specimens with different viscous particle contents decreases with an increase in temperature, except for the specimen with 55.39% viscous particle content; it increased substantially with an increase in temperature. When $\omega = 41\%$, except for the specimen with 45.39% viscous content, the average fissure width of other specimens with different viscous content increased with an increase in temperature. The specimen with 15.39% viscous content showed significant overall shrinkage at a temperature of 60 °C; its fissure width decreased significantly with an increase in temperature. When $\omega = 56.5\%$, the crack width of this specimen increased with an increase in temperature.

### 4.4. Effects of Water Content on Crack Development

In crack development, water content has an obvious influence. An analysis was carried out to investigate the influence of water content on the crack rate, crack length and average crack width, with results shown in Figures 9–11.

**Table 2.** Average crack width.

| Sample Grouping | | | Average Crack Width/mm | Sample Grouping | | | Average Crack Width/mm |
|---|---|---|---|---|---|---|---|
| Water Content | Temperature | Clay Content | | Water Content | Temperature | Clay Content | |
| ω = 24% | 20 °C | 45.39% | 0 | ω = 41% | 20 °C | 45.39% | 2.71 |
| | | 55.39% | 0 | | | 55.39% | 1.29 |
| | | 65.39% | 0 | | | 65.39% | 1.02 |
| | | 75.39% | 0.27 | | | 75.39% | 0.48 |
| | 60 °C | 45.39% | 0 | | 60 °C | 45.39% | 1.20 |
| | | 55.39% | 0.33 | | | 55.39% | 1.33 |
| | | 65.39% | 0 | | | 65.39% | 1.13 |
| | | 75.39% | 0 | | | 75.39% | 0.76 |
| ω = 31.8% | 20 °C | 45.39% | 1.43 | ω = 56.5% | 20 °C | 45.39% | 6.06 |
| | | 55.39% | 1.19 | | | 55.39% | 1.35 |
| | | 65.39% | 0.64 | | | 65.39% | 1.19 |
| | | 75.39% | 0 | | | 75.39% | 0.51 |
| | 60 °C | 45.39% | 1.17 | | 60 °C | 45.39% | 6.06 |
| | | 55.39% | 1.49 | | | 55.39% | 3.01 |
| | | 65.39% | 0.47 | | | 65.39% | 2.38 |
| | | 75.39% | 0 | | | 75.39% | 1.49 |

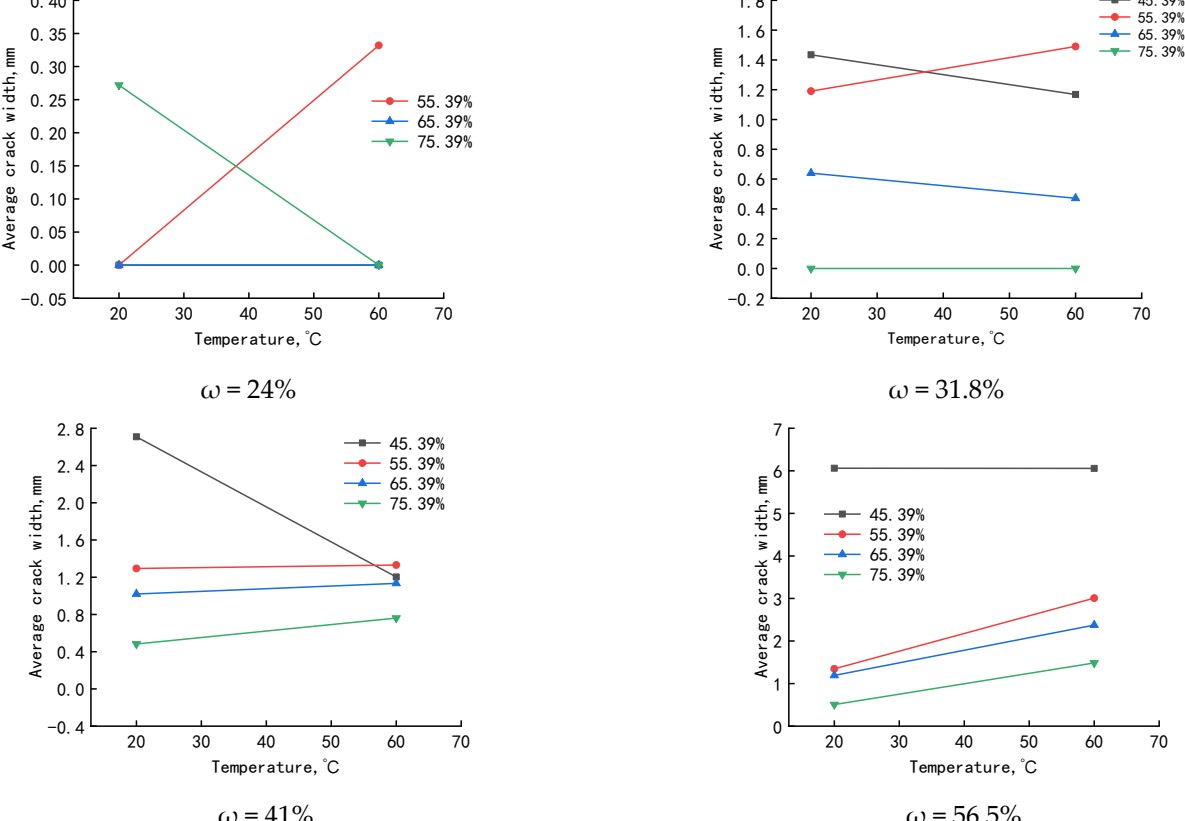

**Figure 8.** Average crack widths with temperature.

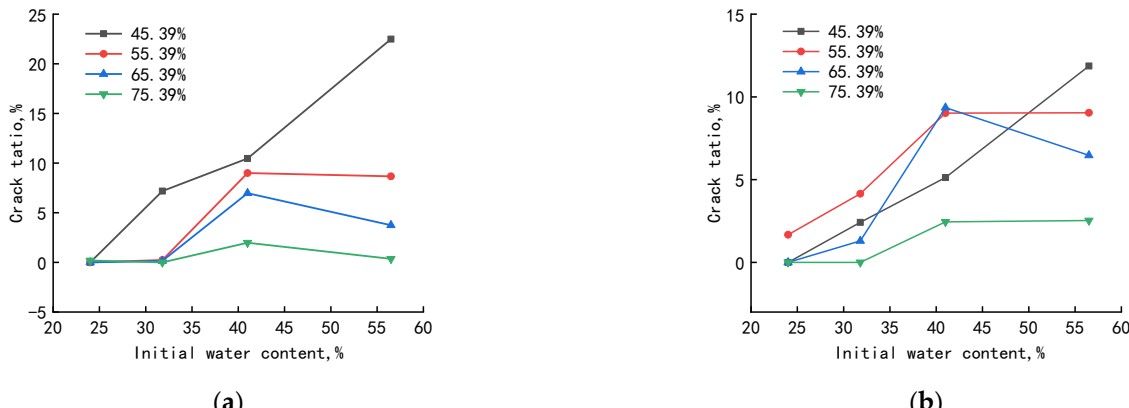

**Figure 9.** Variation curves for crack ratio with water content for (**a**) 20 °C; (**b**) 60 °C.

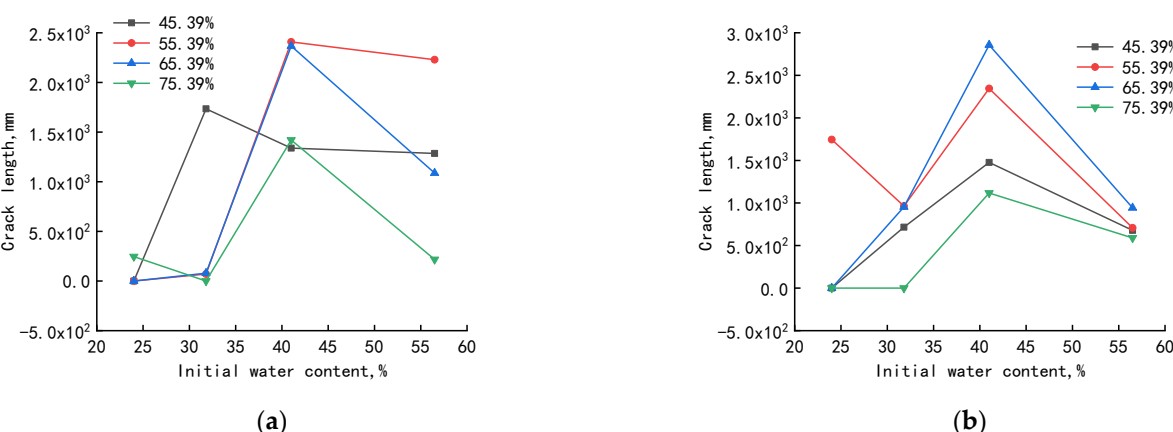

**Figure 10.** Variation curves for crack length with water content for (**a**) 20 °C; (**b**) 60 °C.

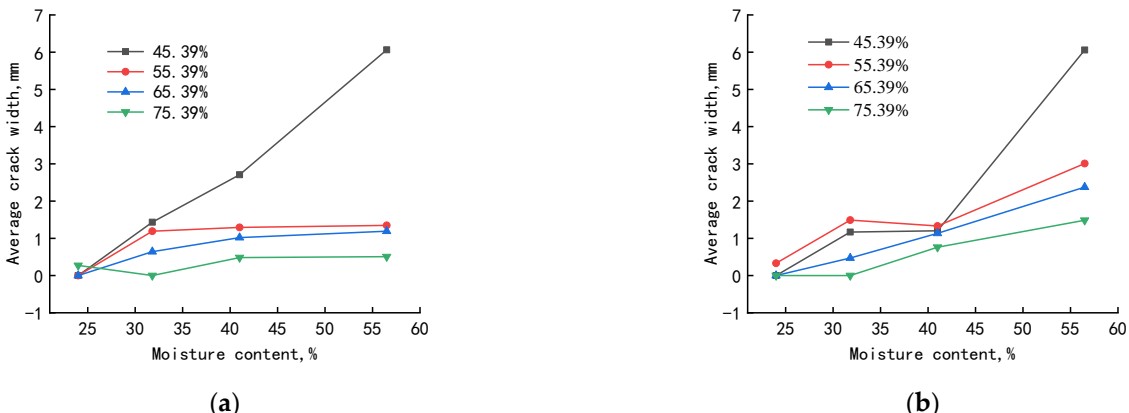

**Figure 11.** Curves for average crack width with water content for (**a**) 20 °C; (**b**) 60 °C.

From Figure 9, it can be seen that with an increase in water content, the fractality of specimens with different viscous particle contents showed an increasing trend at different temperatures. When $\omega$ = 56.5% and the temperature was 20 °C, the crack rate of the specimen with 45.39% mucilage content gradually increased, while the crack rate of other specimens with mucilage content decreased. The crack rates for $\omega$ = 56.5% and $\omega$ = 41% showed that differences decrease with an increase in mucilage content, and decrease to negative when the mucilage content is 65.39% and 75.39%. When the temperature was 60 °C, the crack rate in the specimen with 45.39% of mucilage increased; when the temperature was 60 °C, the fractality of the specimen with 45.39% viscous content increased, and the fractality of specimens with other viscous contents decreased. This phenomenon indicates

that cracking in red clay increases with an increase in clay content and water content, when $\omega > 41\%$.

As can be seen from Figure 10, when the temperature was 20 °C, the water content increased from 31.8% to 56.5%. The fissure lengths in specimens with different viscous particle content increased initially, and then decreased, with the fissure length in specimens with $\omega = 41\%$ being the longest. The fissure length increased initially and then decreased with an increase in the viscous particle content of specimens, with the fissure length of 55.39% viscous particle content specimens being the largest. When the temperature was 60 °C, the moisture content increased from 31.8% to 56.5%, the fissure lengths in specimens with different viscous particle content increased initially and then decreased, with the fissure length in specimens with $\omega = 41\%$ being the longest. The fissure length increased initially and then decreased with an increase in viscous particle content of specimens, with the fissure length in specimens with 65.39% viscous particle content being the largest.

From Figure 11, it can be seen that the crack width in specimens with the same viscous particle content continued to increase with an increase in water content when the temperature was 20 °C. With an increase in viscous particle content, the crack width in specimens with the same water content decreased. At a temperature of 60 °C, with an increase in water content, the crack width in specimens with 45.39% and 55.39% mucilage content showed a small decrease at $\omega = 41\%$, but the overall trend was upward. The crack width in specimens with 65.39% and 75.39% mucilage content increased with an increase in water content. With an increase in mucilage content from 55.39% to 75.39%, the crack width in specimens with the same water content decreased. The crack width of the specimens with the same water content decreased with an increase in water content from 55.39% to 75.39%.

*4.5. Surface Shrinkage in Red Clay with Different Viscous Particle Contents*

During water loss and cracking in red clay, in addition to the fissures generated in the specimen, the soil body also shrunk in the center as a whole, creating a void between the specimen and the mold. Due to surface area limitations of the specimen, this part of shrinkage did not appear on the surface of the specimen as a fissure. However, in engineering practice, this part of the soil shrinkage will also appear in the form of fissures, thus this part of the shrinkage must also be taken into account. The surface shrinkage in red clay specimens with different viscous particle contents is shown in Table 3.

**Table 3.** Surface shrinkage in red clay with different clay contents.

| Clay Content/% | Temperature/°C | Water Content/% | | | |
|---|---|---|---|---|---|
| | | $\omega = 24\%$ | $\omega = 31.8\%$ | $\omega = 41\%$ | $\omega = 56.5\%$ |
| 45.39 | 20 | 4.06 | 13.01 | 17.15 | 22.49 |
| | 60 | 7.21 | 13.09 | 26.34 | 23.53 |
| 55.39 | 20 | 0.04 | 8.34 | 13.36 | 10.91 |
| | 60 | 4.42 | 11.27 | 17.01 | 21.02 |
| 65.39 | 20 | 2.04 | 3.13 | 6.97 | 7.60 |
| | 60 | 1.50 | 4.72 | 14.88 | 13.49 |
| 75.39 | 20 | 0.19 | 0.00 | 1.98 | 4.23 |
| | 60 | 0.00 | 0.88 | 9.73 | 7.36 |

From Table 3, it can be seen that both temperature and water content have an effect on the surface shrinkage in red clay specimens. The relationship between surface shrinkage in specimens with different viscous particle contents and temperature and water content are shown in Figures 12 and 13, respectively.

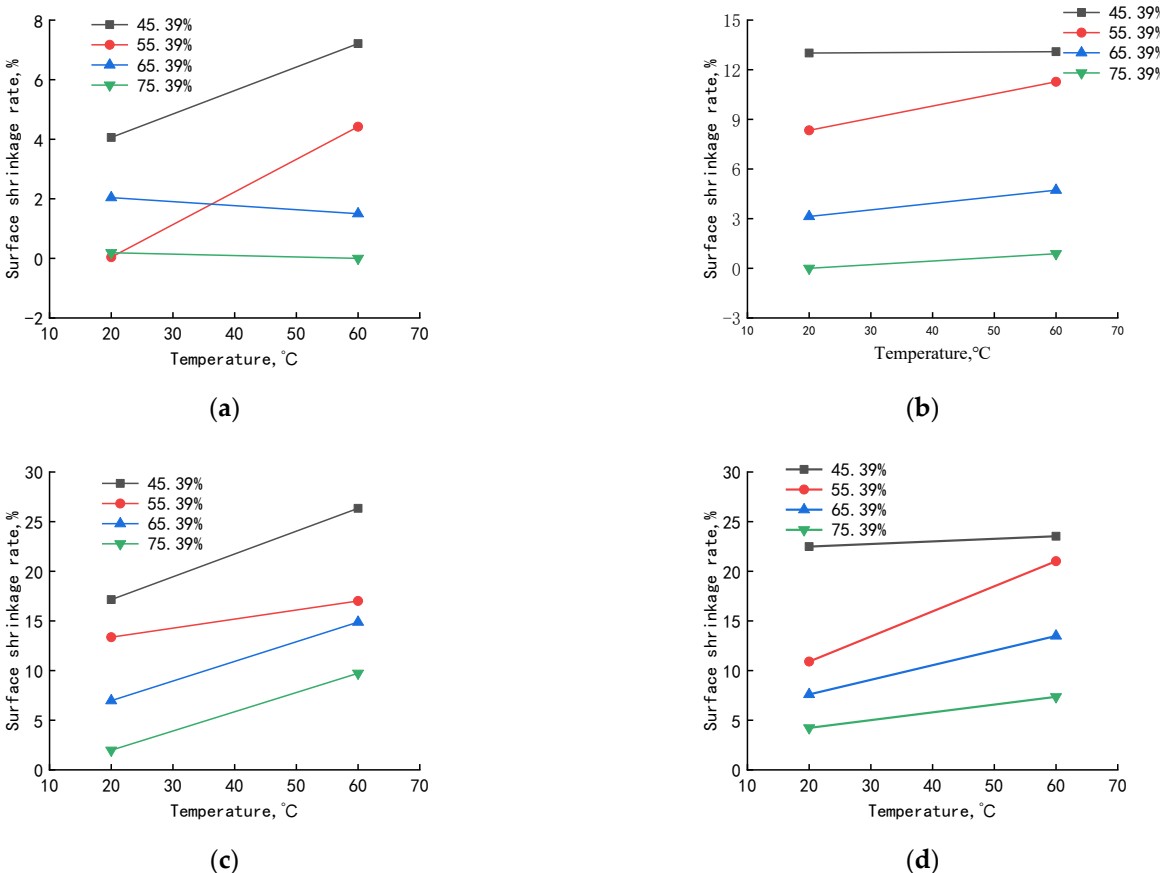

**Figure 12.** Relationships between the surface shrinkage rate and the temperature for (**a**) ω = 24%; (**b**) ω = 31.8%; (**c**) ω = 41%; (**d**) ω = 56.5%.

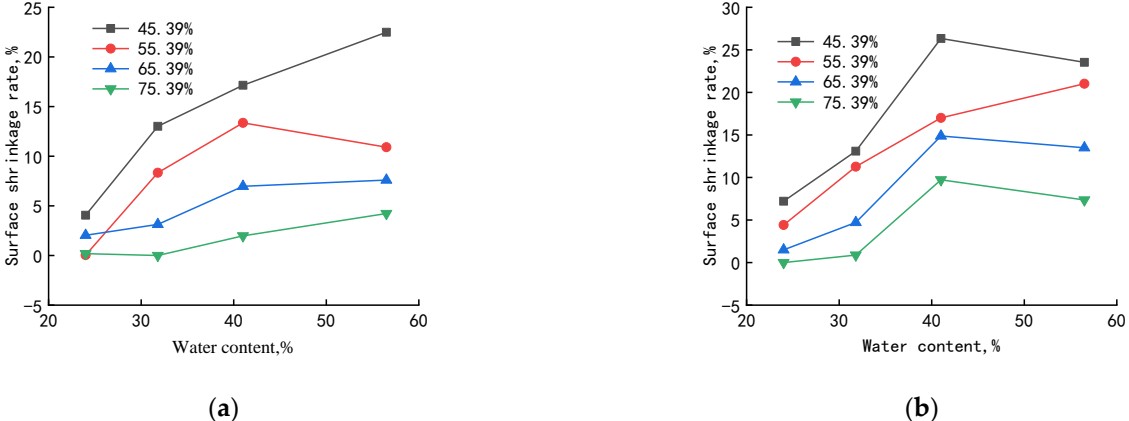

**Figure 13.** Relationships between the surface shrinkage rate and water content at (**a**) 20 °C; (**b**) 60 °C.

As can be seen from Figure 12, when ω = 24%, the surface shrinkage in specimens with 45.39% and 55.39% viscous particles increased with an increase in temperature. The surface shrinkage in specimens with 65.39% and 75.39% viscous particles decreased with an increase in temperature, and the surface shrinkage decreased with an increase in viscous particles in specimens. At other moisture contents, the surface shrinkage in specimens increased with an increase in temperature. The surface shrinkage in specimens with other moisture contents increased with an increase in temperature, and the surface shrinkage in specimens with an increase in viscous particle content decreased.

As can be seen from Figure 13, when the temperature was 20 °C, the surface shrinkage of the specimen with 55.39% viscous particles increased with an increase in water content

and then decreased, with the shrinkage rate being largest when $\omega = 41\%$. The surface shrinkage of the specimens with 45.39%, 65.39% and 75.39% viscous particles increased with an increase in water content, and the surface shrinkage rate decreased with an increase in viscous particles in the specimen. When the temperature was 60 °C, the surface shrinkage rate of specimens with 55.39% viscous particles increased with an increase in water content, and the surface shrinkage rate of specimens with 45.39%, 65.39% and 75.39% viscous particles increased with an increase in water content and then decreased, with the shrinkage rate being largest when $\omega = 41\%$. The surface shrinkage rate decreased with an increase in viscous particle content in specimens.

## 5. Conclusions

The role of mucilage content, temperature and initial water content on the development of cracks in red clay is mainly due to the instability of the interlayer linkage bonds of the lattice in the mucilage minerals; furthermore, the water film intercalation formed by the entry of water molecules causes lattice expansion, leading to macroscopic expansion. In addition, when the soil particles interact with water, the double electric layer ions formed by lattice replacement on the surface adsorb water molecules around the mucilage particles, forming a thin film, which in turn expands the spacing of soil particles. This leads to soil expansion, and with a change in temperature, the thinning and disappearance of the film also causes the soil to contract.

1.  The development of fissures in red clay can be divided into three stages: fissure generation, fissure development and fissure stabilization.
2.  With $\omega = 31.8\%$, 41% and 56.5%, the crack rate in red clay with the same clay grain content increased with an increase in temperature. The fissure length in red clay is related to water content and clay grain content, and no obvious pattern with temperature was found.
3.  There is a connection between temperature, water content and viscous grain content, with the average fissure width in red clay. When the water content is the same, the average fissure width in red clay having the same viscous particle content increased with an increase in water loss temperature; when the temperature was the same, the average fissure width in red clay having the same viscous particle content increased with an increase in water content. When the temperature and water content were the same, the crack width in red clay increased and then decreased with an increase in viscous particle content.
4.  When the water content was the same, surface shrinkage in red clay with the same viscous particle content increased with an increase in water loss temperature. When the water loss temperature was 20 °C, surface shrinkage in red clay with the same viscous particle content increased with an increase in water content; when the water loss temperature was 60 °C, surface shrinkage in red clay with the same viscous particle content increased with an increase in water content, and then decreased.
5.  When $\omega = 41\%$, the surface shrinkage rate in red clay was the largest. When the temperature and water content were the same, surface shrinkage in red clay gradually decreased with an increase in viscous particle content.

The clay content in red clay soil from the Guilin area varies greatly, and the region's climate is hot and rainy in the summer. According to the test results of this paper, when the surface temperature is greater than 30 °C and the water content is greater than 31.8%, fissures in red clay develop strongly. Thus, in engineering practices, attention should be paid to reducing surface temperatures, controlling the water content of the surface red clay, and to ensure sound drainage protection.

**Author Contributions:** Conceptualization, B.L. and B.Y.; methodology, B.Y.; software, L.W.; validation, B.L. and L.W.; formal analysis, L.W.; investigation, B.Y.; resources, B.L.; data curation, L.W.; writing—original draft preparation, L.W.; writing—review and editing, L.W. and B.Y.; visualization,

L.W.; supervision, B.L.; project administration, B.L.; funding acquisition, B.L. All authors have read and agreed to the published version of the manuscript.

**Funding:** This research was funded by [the National Natural Science Foundation of China] grant number [42067044]; [projection of Guangxi Key Laboratory, building new energy and energy conservation] grant number [16-j-21-6, 19-j-21-21].

**Institutional Review Board Statement:** Not applicable.

**Informed Consent Statement:** Not applicable.

**Data Availability Statement:** Data are contained within the paper.

**Conflicts of Interest:** The authors declare no conflict of interest.

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
