# Peer review of "Study on Crack Development in Red Clay from Guangxi Guilin with Different Clay Grain Content"

_sustainability, doi:10.3390/su142013104_

Round 1
Reviewer 1 Report
This is a nicely framed and executed paper. The topic is interesting and the paper provides original results based on a pragmatic approach however there are several issues that need to be addressed:
1. The first sentence of the abstract is very long. The author should consider splitting it.
2. The description of the experimental setup is too synthetic. The main issue is that the boundary conditions are not clearly described. Are the clay specimens still confined? Of course, the loading of the lateral boundary of the specimen could have a significant effect on crack onset, growth and opening. This point should be discussed in the paper.
3. The expressions “crack width” and “crack area” are not clearly defined, please described what are those variables.
4. There are numerous formatting problems in the bibliography, the authors should correct this.
Reviewer 2 Report
In this manuscript, four specimens with different water contents were prepared, and the fracture development patterns of the specimens were observed under two temperature conditions of 20℃ and 60℃, and the fracture area, fracture rate, fracture length and width were measured and calculated by using Image J image processing software. The study is interesting, and this manuscript can be accepted with minor modifications.
1. To ensure the rigor of this manuscript, some references need to be supplemented, such as line 395, 397, 403, 409, etc. In addition, it is recommended to effectively quote the relevant literature of this Journal in recent years.
2. For the clay particles mentioned in this manuscript, the specific preparation process should be given.
3. The thickness also has a certain influence on the development of fractures. Why is the thickness not considered during the test in this manuscript?
4. Fonts in figures need to adjust to make it clearer. Such as Figure 1(a), Figure 13, etc. Some titles need to unify, such as ‘Fig.’ in line 66 and ‘Figure’ in line 112.
